# A density estimation perspective on learning from pairwise human preferences

**Vincent Dumoulin**[†], **Daniel D. Johnson**[†‡], **Pablo Samuel Castro**[†§], **Hugo Larochelle**[†§¶],
**Yann Dauphin**[†]
*{vdumoulin,ddjohnson,psc,hugolarochelle,ynd}@google.com*
[†] *Google DeepMind*
[‡] *University of Toronto*
[§] *Mila, Université de Montréal*
[¶] *CIFAR Fellow*

**Reviewed on OpenReview:** *https://openreview.net/forum?id=YH3oERVYjF*

## Abstract

Learning from human feedback (LHF)—and in particular learning from pairwise preferences—has recently become a crucial ingredient in training large language models (LLMs), and has been the subject of much research. Most recent works frame it as a reinforcement learning problem, where a reward function is learned from pairwise preference data and the LLM is treated as a policy which is adapted to maximize the rewards, often under additional regularization constraints. We propose an alternative interpretation which centers on the generative process for pairwise preferences and treats LHF as a density estimation problem. We provide theoretical and empirical results showing that for a family of generative processes defined via preference behavior distribution equations, training a reward function on pairwise preferences effectively models an annotator's implicit preference distribution. Finally, we discuss and present findings on "annotator misspecification"—failure cases where wrong modeling assumptions are made about annotator behavior, resulting in poorly-adapted models—suggesting that approaches that learn from pairwise human preferences could have trouble learning from a population of annotators with diverse viewpoints.[1]

## 1 Introduction

With the recent surge of interest in large language models (LLMs), learning from human feedback (LHF) has received a great deal of attention from the machine learning community. Pretraining of large Transformer architectures (Vaswani et al., 2017) on web-scale data has been essential to their success, but by itself the pretraining objective produces unconditional generative models that are difficult to control. A critical component in training LLMs is therefore to finetune them to produce "good" responses, which is typically formalized as aligning their outputs with human preferences.

The current dominant approach to aligning LLMs with human preferences relies on pairwise comparisons of model outputs, and frames the problem as a reinforcement learning (RL) problem where the LLM is treated as a policy. Preference-based learning of reward functions (Sadigh et al., 2017; Biyik & Sadigh, 2018)—and more specifically preference-based RL or RL from human preferences (abbreviated as RLHF; Wirth et al., 2017; Christiano et al., 2017; Ibarz et al., 2018; Brown et al., 2019; 2020; Lee et al., 2021; Shin et al., 2023)—is used to finetune LLMs from model output pairs which have been ranked by annotators to reflect how preferable those outputs are (Stiennon et al., 2020; Ouyang et al., 2022; Bai et al., 2022).

However, interpreting preference-alignment as an RL problem requires a change of perspective from the original pretraining task. For LLM pretraining, the loss is derived from a probabilistic modeling formulation

---

[1] A notebook reproducing all experiments in this paper can be accessed at `https://github.com/google-deepmind/pbde`.

of the problem: The language model is treated as an *autoregressive probability distribution*, and next-token prediction is used to maximize the likelihood of human language samples. For human preference alignment, the language model is treated as a *policy* and the loss aims to maximize a (learned) reward which is reflective of human preferences. Given the probabilistic origins of training LLMs, it is worth considering whether certain probabilistic modeling tools and techniques are being overlooked when taking the RL perspective.

We argue that treating human preference alignment as a probabilistic modeling task forces us to be explicit about our assumptions on the *generative process* for annotator preferences and provides important insights into the properties and failure modes of algorithms that learn from them. Our main contributions are:

- We show that the standard procedure for training a reward function on pairwise human preferences can be reinterpreted as performing density estimation on an annotator's *implicit preference distribution*, assuming annotators behave according to the *Luce choice rule* (Luce et al., 1963);

- We generalize the Luce choice rule to a family of generative processes for pairwise preferences defined via *preference behavior distribution equations* (PBDEs) and provide global optimality results in the case where the annotator and model behave according to the same PBDE;

- Finally, we show that an unintentional mismatch in generative processes for pairwise preferences between the annotator and model (which we term *annotator misspecification*) can lead to badly-tuned policies, highlighting the importance of explicitly stating assumptions on annotator behavior.

## 2 Background

Here we provide some technical background and notation which will be used throughout the manuscript.

### 2.1 Reinforcement Learning

Reinforcement learning tackles sequential decision-making problems and is typically formulated as a Markov decision process $\mathcal{M} = \{\mathcal{X}, \mathcal{A}, \mathcal{P}, \mathcal{R}, \gamma\}$ using the following definitions: $\mathcal{X}$ is the set of states; $\mathcal{A}$ is a set of actions (or "one-step decisions"); $\mathcal{P} : \mathcal{X} \times \mathcal{A} \to \Delta(\mathcal{X})$ defines the transition dynamics, where $\mathcal{P}(x, a)(x')$ is the probability of transitioning to state $x'$ after performing action $a$ from state $x$; $\mathcal{R} : \mathcal{X} \times \mathcal{A} \to \mathbb{R}$ is the one-step reward, where $\mathcal{R}(x, a)$ is the reward received after performing action $a$ from state $x$; $\gamma \in [0, 1)$ is a discount factor that places more emphasis on rewards received in the short-term.

An agent's behaviour is formalized via a policy $\pi : \mathcal{X} \to \Delta(\mathcal{A})$ which gives a distribution over possible actions conditioned on a state (e.g., $\pi(x)(a)$ is the probability given by policy $\pi$ of picking action $a$ when in state $x$). Given a fixed policy $\pi$, we can express rewards as a function of state: $r_\pi(x) := \mathbb{E}_{a \sim \pi(x)} \mathcal{R}(x, a)$; this notation will prove useful when considering reward functions over LLM outputs. Every policy $\pi$ induces a *value function* given by the expected sum of discounted rewards when following policy $\pi$:

$$V^\pi(x) = \sum_{t=0}^{\infty} \left[ \gamma^t \mathcal{R}(x_t, a_t) \mid x_0 = x, a_t \sim \pi(x_t), x_{t+1} \sim \mathcal{P}(x_t, a_t) \right].$$

It is useful to also consider state-action value functions, which quantify the value of taking an action $a$ from state $x$, and following the policy $\pi$ afterwards: $Q^\pi(x, a) := \mathcal{R}(x, a) + \gamma \mathbb{E}_{x' \sim \mathcal{P}(x, a)} V^\pi(x')$. Finally, we denote by $\mu_\pi \in \Delta(\mathcal{X})$ the stationary state distribution induced by a policy $\pi$.

The goal of RL agents is then to find the optimal policy $\pi^*$, where optimality is defined uniformly across the state space (e.g. $V^{\pi^*} \geq V^\pi$ for every $\pi$). In most modern RL applications, policies are expressed by neural networks with parameters $\theta$, and we denote the respective policy as $\pi_\theta$. There are a number of approaches for tackling this problem (Sutton & Barto, 1998), but for the purposes of this work we focus on policy gradient methods. Specifically, the *policy gradient Theorem* asserts the following relationship:

$$\nabla V^{\pi_\theta} \propto \sum_{x \in \mathcal{X}} \mu_{\pi_\theta}(x) \sum_{a \in \mathcal{A}} Q^{\pi_\theta}(x, a) \nabla \pi_\theta(x)(a).$$

This result establishes that we can use gradient-based optimization methods to maximize returns by only requiring gradients of the parameterized policy $\pi_\theta$. Therefore we can consider the output of any neural network (such as an LLM) as a policy, and we can then update this policy so as to maximize some reward function of interest. This observation is what underlies RLHF, which we discuss in the next section.

## 2.2 Reinforcement Learning from Human Feedback (RLHF)

RLHF works in two stages: (1) a reward function is learned from the annotator preference data; and (2) the LLM is treated as an RL policy and finetuned to maximize the learned reward function. The two stages are usually alternated in multiple iterations of a closed loop where new annotator feedback is gathered between each iteration.

More formally, the annotator ranks pairs of model outputs $\boldsymbol{x}_A$ and $\boldsymbol{x}_B$, and the outcome can be represented with a binary variable $y$ that takes the value $y = 1$ if $\boldsymbol{x}_A$ is preferred to $\boldsymbol{x}_B$ (noted $\boldsymbol{x}_A \succ \boldsymbol{x}_B$) and $y = 0$ otherwise. In the context of language modeling, the reward function is usually defined as a parameterized mapping $r_\phi(\boldsymbol{x}_0, \boldsymbol{x})$ from prompt $\boldsymbol{x}_0$ and continuation $\boldsymbol{x}$ to a scalar reward. To simplify the notation and treatment, we will omit $\boldsymbol{x}_0$ and consider the unconditional generation case, with the understanding that the discussion extends trivially to the conditional generation case. The reward function $r_\phi(\boldsymbol{x})$ is trained on a dataset $D$ of annotator preferences:

$$\phi \leftarrow \min_\phi \quad \mathbb{E}_{\boldsymbol{x}_A, \boldsymbol{x}_B, y \sim D}\bigg[ -y \cdot \log p_\phi(y; \boldsymbol{x}_A, \boldsymbol{x}_B) - (1 - y) \cdot \log(1 - p_\phi(y; \boldsymbol{x}_A, \boldsymbol{x}_B)) \bigg] \tag{1}$$

where

$$p_\phi(y; \boldsymbol{x}_A, \boldsymbol{x}_B) = \sigma\Big( r_\phi(\boldsymbol{x}_A) - r_\phi(\boldsymbol{x}_B) \Big) = \frac{e^{r_\phi(\boldsymbol{x}_A)}}{e^{r_\phi(\boldsymbol{x}_A)} + e^{r_\phi(\boldsymbol{x}_B)}} \tag{2}$$

is the probability that $\boldsymbol{x}_A \succ \boldsymbol{x}_B$ under the Bradley-Terry model (Bradley & Terry, 1952) for pairwise comparisons. In plain language, the reward function is trained using a binary cross-entropy loss on the comparison outcome variable $y$, which incentivizes increasing the reward margin between the preferred and non-preferred outputs. Then, the LLM (noted $\pi_\theta(\boldsymbol{x})$)[2] is tuned so that

$$\theta \leftarrow \max_\theta \quad \mathbb{E}_{\pi_\theta(\boldsymbol{x})}\bigg[ r_\phi(\boldsymbol{x}) \bigg]. \tag{3}$$

Using the well-known identity for policy gradient in RL, this results in a gradient of the form

$$\nabla_\theta \mathbb{E}_{\pi_\theta(\boldsymbol{x})}\bigg[ r_\phi(\boldsymbol{x}) \bigg] = \mathbb{E}_{\pi_\theta(\boldsymbol{x})}\bigg[ r_\phi(\boldsymbol{x}) \nabla_\theta \log \pi_\theta(\boldsymbol{x}) \bigg]. \tag{4}$$

In order to prevent "catastrophic forgetting" in the model, it is common to combine the learned reward with a KL-divergence term between the finetuned LLM and the pretrained LLM (noted $\pi_{\text{pre}}$)—a practice known as *KL-control*—yielding a gradient of the form

$$\mathbb{E}_{\pi_\theta(\boldsymbol{x})}\bigg[ \bigg( r_\phi(\boldsymbol{x}) - \beta \log \frac{\pi_\theta(\boldsymbol{x})}{\pi_{\text{pre}}(\boldsymbol{x})} \bigg) \nabla_\theta \log \pi_\theta(\boldsymbol{x}) \bigg]. \tag{5}$$

Such a constraint is usually handled using policy gradient methods such as proximal policy optimization (PPO; Schulman et al., 2017).

---

[2]The choice of notation stems from the fact that it is treated as a policy for the purpose of RLHF.

## 3 Related work

Our work is related to numerous previous works investigating alternatives to RL to learn from pairwise human preferences.

Liu et al. (2023a) move away from RL and propose a technique called Chain of Hindsight which turns annotator-ranked pairs of model outputs and their associated contexts into training sequences of the form "{Context} {Positive prefix}: {Desirable output} {Negative prefix}: {Undesirable output}". The model is finetuned into these chain-of-hindsight sequences and then conditioned at inference time by appending "{Positive prefix}" to the annotator prompt.

Korbak et al. (2022) examine the relationship between reward maximization approaches like RLHF (with or without KL-control) and distribution matching approaches like distributional policy gradients (DPG). They show that RLHF with KL-control can also be thought of as a distribution matching approach, as it is equivalent to minimizing the reverse KL-divergence between $\pi_\theta(\boldsymbol{x})$ and an energy-based model of the form $p_z(\boldsymbol{x}) \propto \pi_{\mathrm{pre}}(\boldsymbol{x})e^{r(\boldsymbol{x})/\beta}$. Our work complements their analysis by characterizing what the optimal reward function captures about the annotator (their implicit preference distribution) under various hypothesized generative processes for pairwise preferences.

Yuan et al. (2023) align the model with pairwise human preferences through a combination of a ranking loss on the log-likelihood margin between preferred and non-preferred model outputs and a negative log-likelihood loss on the preferred model output. Zhao et al. (2023) optimize for pairwise preference alignment using a hinge loss on the log-likelihood margin between preferred and non-preferred model outputs and a regularization term towards a model finetuned with supervision. Liu et al. (2023b) extend the loss introduced by Zhao et al. (2023) by normalizing the log-likelihood margin using $\pi_{\mathrm{pre}}(\boldsymbol{x})$. However, their primary contribution is a statistical rejection sampling algorithm called RSO that draws samples from the optimal policy using samples from $\pi_{\mathrm{pre}}(\boldsymbol{x})$ and a learned reward function.

Rafailov et al. (2023) show an equivalence between RLHF with KL-control and binary classification of pairwise comparison outcomes when the reward function used to make a prediction (Equation 2) is defined as $r(\boldsymbol{x}) = \beta(\log \pi_\theta(\boldsymbol{x}) - \log \pi_{\mathrm{pre}}(\boldsymbol{x}))$ (called DPO, for direct preference optimization). Azar et al. (2023) frame DPO as a special case of a broader family of approaches called $\Psi$PO where the maximized quantity is a function $\Psi$ of the preference probability defined by the Bradley-Terry model. They introduce another special case called identity preference optimization (IPO) which is optimized via a quadratic loss instead of a sigmoid binary cross-entropy loss and is meant to make KL-regularization more effective in the face of deterministic annotator preferences. Whereas Azar et al. (2023) consider various transformations of the preference probability for the optimal reward under the Bradley-Terry model, we systematically use the logit function on the preference probability and instead examine various transformations of $\log \pi_\theta(\boldsymbol{x})$ to obtain the preference probability itself.

## 4 A probabilistic interpretation of learning from pairwise human preferences

We now consider a probabilistic interpretation of LHF in the case where the training signal originates from pairwise preferences. First, we need to consider the generative process for the preferences and make explicit assumptions about it.

### 4.1 Reward learning as density estimation under the Luce choice rule

Let $p^*(y; \boldsymbol{x}_A, \boldsymbol{x}_B)$ be an annotator's preference probability whose functional form we will make more explicit shortly and encapsulates our assumptions on the generative process for pairwise human preferences. Let $q(\boldsymbol{x}_A, \boldsymbol{x}_B)$ be a *proposal distribution* over pairs $\boldsymbol{x}_A$ and $\boldsymbol{x}_B$ to be ranked by the annotator. One common choice of proposal is to draw independent sample pairs from $\pi_{\mathrm{pre}}(\boldsymbol{x})$. Using the two distributions we can decompose the true distribution $D$ of preference data used in the expectation in Equation 1 into

$$D(\boldsymbol{x}_A, \boldsymbol{x}_B, y) = q(\boldsymbol{x}_A, \boldsymbol{x}_B)p^*(y; \boldsymbol{x}_A, \boldsymbol{x}_B), \qquad (6)$$

and we can rewrite Equation 1 as

$$\phi \leftarrow \min_{\phi} \quad \mathbb{E}_{q(\boldsymbol{x}_A, \boldsymbol{x}_B)} \left[ \mathbb{E}_{p^*(y; \boldsymbol{x}_A, \boldsymbol{x}_B)} \left[ -y \cdot \log p_{\phi}(y; \boldsymbol{x}_A, \boldsymbol{x}_B) - (1 - y) \cdot \log(1 - p_{\phi}(y; \boldsymbol{x}_A, \boldsymbol{x}_B)) \right] \right], \quad (7)$$

which is equivalent[3] to

$$\phi \leftarrow \min_{\phi} \quad \mathbb{E}_{\boldsymbol{x}_A, \boldsymbol{x}_B \sim q(\boldsymbol{x}_A, \boldsymbol{x}_B)} \left[ D_{\mathrm{KL}} \big( p^*(y; \boldsymbol{x}_A, \boldsymbol{x}_B) \,||\, p_{\phi}(y; \boldsymbol{x}_A, \boldsymbol{x}_B) \big) \right] \quad (8)$$

and is globally minimized when the KL-divergence is zero for all possible $\boldsymbol{x}_A$ and $\boldsymbol{x}_B$ pairs.

The correct assumption to make about the generative process for pairwise human preferences is essentially an empirical question, but a reasonable hypothesis is a model of decision processes known as the Luce choice rule (or ratio rule; Shepard, 1957; Luce et al., 1963). As Sanborn & Griffiths (2007) and Peterson et al. (2018) explain, a common assumption about the behavior of rational Bayesian subjects is that they make choices according to some *implicit preference distribution $p^*(\boldsymbol{x})$* over outcomes: for any two outcomes $\boldsymbol{x}_A$ and $\boldsymbol{x}_B$, they are assumed to prefer $\boldsymbol{x}_A$ over $\boldsymbol{x}_B$ with probability

$$\mathrm{Prob}(\boldsymbol{x}_A \succ \boldsymbol{x}_B) = \frac{p^*(\boldsymbol{x}_A)}{p^*(\boldsymbol{x}_A) + p^*(\boldsymbol{x}_B)}. \quad (9)$$

This model is also known as the Bradley-Terry model (Bradley & Terry, 1952). In fact, Sanborn & Griffiths (2007) and Peterson et al. (2018) use this very assumption to place human subjects at the core of a Monte Carlo Markov Chain (MCMC) by having them perform the accept/reject step and ultimately drawing samples from their implicit preference distribution $p^*(\boldsymbol{x})$ over outcomes.

Equation 9 and Equation 2 share the same functional form for a reason: if we assume the generative process for pairwise human preferences follows the Luce choice rule, then predicting pairwise comparison outcomes using the same generative process results in a well-specified model of annotator behavior.

**Theorem 1.** *Let $p^*(\boldsymbol{x})$ be a probability distribution with support $\mathcal{S}$, and let $q(\boldsymbol{x}_A, \boldsymbol{x}_B)$ be a joint probability distribution with support $\mathcal{S} \times \mathcal{S}$. Assume $q(\boldsymbol{x}_A, \boldsymbol{x}_B) > 0$ for all $\boldsymbol{x}_A, \boldsymbol{x}_B \in \mathcal{S} \times \mathcal{S}$. Let $p_{\phi}(y; \boldsymbol{x}_A, \boldsymbol{x}_B)$ and $p^*(y; \boldsymbol{x}_A, \boldsymbol{x}_B)$ be defined according to Equations 2 and 9, respectively. The loss function*

$$\mathbb{E}_{\boldsymbol{x}_A, \boldsymbol{x}_B \sim q(\boldsymbol{x}_A, \boldsymbol{x}_B)} \left[ D_{\mathrm{KL}} \big( p^*(y; \boldsymbol{x}_A, \boldsymbol{x}_B) \,||\, p_{\phi}(y; \boldsymbol{x}_A, \boldsymbol{x}_B) \big) \right]$$

*is globally minimized when*

$$e^{r_{\phi}(\boldsymbol{x})} \propto p^*(\boldsymbol{x}), \quad \forall \boldsymbol{x} \in \mathcal{S}.$$

A proof is provided in Appendix A.1. We can therefore reinterpret the training of a reward function on pairwise human preferences as the following probabilistic procedure:

1. We assume the generative process for pairwise human preferences follows the Luce choice rule (Equation 9) and consider annotator preferences as samples from this process;

2. We predict comparison outcomes with the learnable reward function using the same generative process (Equation 2);

3. We use a maximum-likelihood criterion to optimize the reward-derived preference probability (Equation 8) and backpropagate the binary cross-entropy loss through Equation 2 and into the reward function itself; and

---

[3]Recall that the entropy of $p^*(\cdot)$ does not depend on $\phi$

4. Provided the Luce choice rule assumption made in step 1 holds, **the resulting optimal reward function is then equal to the annotator's implicit log-preference distribution** (up to some additive constant).

To demonstrate this, consider a univariate implicit preference distribution

$$p^*(x) = \frac{2}{5} \cdot N_{\text{truncated}}(\mu = -2.5, \sigma = 0.25) + \frac{3}{5} \cdot N_{\text{truncated}}(\mu = 2.5, \sigma = 1.0) \tag{10}$$

defined over the $[-10, 10]$ interval (Figure 1, dashed blue). We simulate learning from pairwise preferences by drawing $2^{15}$ observation pairs $\boldsymbol{x}_A$ and $\boldsymbol{x}_B$ uniformly at random in $[-10, 10] \times [-10, 10]$ and drawing corresponding comparison outcomes $y$ using Equation 9. We then train a reward function in the form of a four-layer MLP (refer to Table 1 for details). Finally, we approximate the normalizing constant for the distribution $p_\phi(\boldsymbol{x}) = e^{r_\phi(\boldsymbol{x})}/Z(\phi)$ and visualize $p_\phi(\boldsymbol{x})$ (Figure 1, solid green). We can clearly see that in the large-data regime (in terms of number of ranked pairs), the reward function learns to model the implicit preference distribution (see also Figure 7 in the Appendix).

Table 1: Univariate toy experiment hyperparameters.

| Hyperparameter | Value |
|---|---|
| Architecture | MLP |
| Hidden layers | 4 layers of width 64 |
| Activation | $tanh$ |
| Optimizer | Adam (Kingma & Ba, 2015) |
| Optimization steps | 8192 |
| Learning rate | $5 \times 10^{-4}$ |
| Learning rate schedule | Cosine decay to 0.0 over 8192 steps |

In other words, if "your language model is secretly a reward model" (Rafailov et al., 2023), it is perhaps equally fair to say that **your reward model secretly learns the annotator's implicit preference distribution**.[4] Ultimately, the reward function is a means to the end of aligning a generative model (for example, an LLM "policy") with human preferences, but the above discussion shows that under the Luce choice rule assumption the optimal reward function actually learns the implicit preference distribution $p^*(\boldsymbol{x})$.

### 4.2 Specifying policies as normalized preference distributions

How, then, should we use our model of human preferences to construct a generative model that aligns with those preferences? We could use an RL algorithm to search for a policy that optimizes preferences, but Theorem 1 suggests that this may not be necessary: the "reward model" can already be interpreted as a probabilistic model of preferred outputs, since it estimates the implicit preference distribution $p^*(\boldsymbol{x})$ that is assumed to govern the annotator's behavior. This means that we could in principle simply use our approximation of the annotator's implicit preference distribution as a policy. The only remaining difficulty is that Theorem 1 only specifies the preference distribution up to a constant.

To address this, suppose we simply define the "reward" $r_\theta(\boldsymbol{x})$ as the log-density of a parameterized generative model, i.e. $r_\theta(\boldsymbol{x}) = \log \pi_\theta(\boldsymbol{x})$.[5] Since $\pi_\theta(\boldsymbol{x})$ is a normalized probability distribution, attaining the global minimum of the loss function **directly** results in $\pi_\theta(\boldsymbol{x}) = p^*(\boldsymbol{x})$. This removes the need for a separate reward model, and the original "reward" is now just implicitly encoded in our learned approximation of the implicit preference distribution.

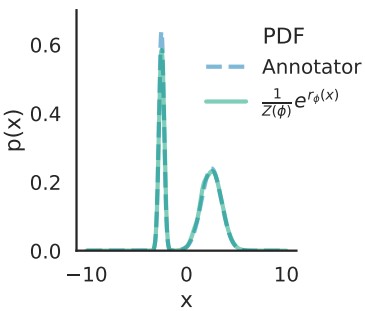

Figure 1: Training a reward model on comparison outcomes stemming from a synthetic implicit preference distribution (Equation 10; dashed blue) recovers the implicit distribution (solid green).

---

[4]Provided the annotator's preferences follow the Luce choice rule.
[5]We purposefully switch from $r_\phi(\boldsymbol{x})$ to $r_\theta(\boldsymbol{x})$ to highlight the fact that the reward function now depends on the LLM's parameters $\theta$.

This idea of bypassing the reward function and directly optimizing the LLM on pairwise human feedback is explored in recent works like ILHF (Xu et al., 2023) and DPO (Rafailov et al., 2023) (which we will discuss in more detail shortly) as well as IPO (Azar et al., 2023), RRHF (Yuan et al., 2023), SLiC-HF (Zhao et al., 2023), and RSO (Liu et al., 2023b) (discussed in Appendix A.2). We also note the probabilistic procedure described above can be thought of as an inductive bias in and of itself: if one believes that the implicit preference distribution is the ideal generative model, then expressing the reward as the LLM's log-likelihood is the correct thing to do. In practice, however, stronger inductive biases are used to regularize the tuned LLM towards its pretrained parameterization.

### 4.3 Expanding to a broader family of generative processes for pairwise preferences

Our results show that—assuming the generative process for the annotator's pairwise preferences follows the Luce choice rule—fitting a model to those preferences using the parameterization in Equation 2 recovers the annotator's implicit preference distribution $p^*(\boldsymbol{x})$. This recasts the standard reward modeling procedure as a density estimation procedure. However, it is not necessarily obvious that annotators exactly follow the Luce choice rule, nor that the implicit preference distribution $p^*(\boldsymbol{x})$ we recover is itself a good "policy".

Luckily, our density estimation results can be extended to hold for generative processes beyond the Luce choice rule. We focus on the following family of equations for describing generative processes which we name **preference behavior distribution equations (PBDEs)**, noted $\Omega_p(\boldsymbol{x})$:

$$p(y; \boldsymbol{x}_A, \boldsymbol{x}_B) = \sigma\Big(\Omega_p(\boldsymbol{x}_A) - \Omega_p(\boldsymbol{x}_B)\Big), \quad \Omega_p(\boldsymbol{x}) = f(\boldsymbol{x}) \cdot \log p(\boldsymbol{x}) + g(x), \quad f(\boldsymbol{x}) > 0 \quad \forall \boldsymbol{x}. \qquad (11)$$

**Theorem 2.** *Let the generative processes for pairwise preferences for the annotator and the model be governed by the **same** PBDE, i.e.,*

$$\Omega_{p^*}(\boldsymbol{x}) = f(\boldsymbol{x}) \cdot \log p^*(\boldsymbol{x}) + g(\boldsymbol{x}) \quad \text{and} \quad \Omega_{\pi_\theta}(\boldsymbol{x}) = f(\boldsymbol{x}) \cdot \log \pi_\theta(\boldsymbol{x}) + g(\boldsymbol{x}).$$

*Then the global minimizer of*

$$\mathbb{E}_{q(\boldsymbol{x}_A, \boldsymbol{x}_B)} \Big[ D_{\mathrm{KL}}\Big( p^*(y; \boldsymbol{x}_A, \boldsymbol{x}_B) \,\|\, p_\theta(y; \boldsymbol{x}_A, \boldsymbol{x}_B) \Big) \Big]$$

*is*

$$\pi_\theta(\boldsymbol{x}) = p^*(\boldsymbol{x}), \quad \forall \boldsymbol{x}.$$

A proof is provided in Appendix A.1. a simple choice of PBDE is the Luce choice rule itself, as we discuss in subsection 4.2. This corresponds to setting $f(\boldsymbol{x}) = 1$ and $g(\boldsymbol{x}) = 0$, and is adopted in practice by the **Inclusive Learning From Human Feedback** (ILHF; Xu et al., 2023) approach.

We now give a concrete example of an alternative PBDE to the Luce choice rule, which stems from the latter's behavior in the context of variable-length inputs like text sequences. In that case, since the likelihood assigned by the implicit preference distribution to a token sequence $\boldsymbol{x}$ is decomposed into a product of per-token likelihoods in $[0, 1]^{|\boldsymbol{x}|}$, longer sequences naturally have lower likelihood magnitudes. This means that an implicit preference distribution represented by an autoregressive model trained only on long sequences could in some cases assign a higher likelihood to short sequences despite having a near-zero probability of sampling them. In other words, if we were to sample from such an implicit preference distribution we would get long sequences, yet using the same distribution to sample preferences with the Luce choice rule would tend to prefer shorter sequences. Refer to subsection 5.2 for a practical illustration of this phenomenon. We argue that this is unrealistic: an annotator should not prefer shorter sequences simply because the total number of possible short sequences is smaller than the total number of possible long sequences. (In fact, real-world human feedback datasets tend to show a slight preference towards *longer* responses (Singhal et al., 2023).)

A more realistic alternative to the Luce choice rule in this context would be the length-normalized variant

$$\text{Prob}(\boldsymbol{x}_A \succ \boldsymbol{x}_B) = \frac{p^*(\boldsymbol{x}_A)^{\frac{1}{|\boldsymbol{x}_A|}}}{p^*(\boldsymbol{x}_A)^{\frac{1}{|\boldsymbol{x}_A|}} + p^*(\boldsymbol{x}_B)^{\frac{1}{|\boldsymbol{x}_B|}}}. \tag{12}$$

The above has the advantage that its outcome is no longer biased by the inherent differences in likelihood magnitudes between sequences of varying lengths. This is likely why length-normalized log-likelihoods have been used by Yuan et al. (2023) in RRHF, for instance. Since Equation 12 is a PBDE with $f(\boldsymbol{x}) = |\boldsymbol{x}|^{-1}$ and $g(\boldsymbol{x}) = 0$, we know through Theorem 2 that it is possible to recover a globally optimal $\pi_\theta(\boldsymbol{x}) = p^*(\boldsymbol{x})$ when both the annotator and the model preferences are sampled according to that generative process.

Theorem 2 provides a global optimality result for a broad family of generative processes for pairwise preferences when the annotator and model share the same process, but it can sometimes be useful to define a different generative process for the model than the one assumed for the annotator—for instance to influence the properties of the globally optimal $\pi_\theta(\boldsymbol{x})$. We now show how **Direct Preference Optimization** (DPO; Rafailov et al., 2023) can be recast as an instance of this general density estimation procedure. DPO bypasses the policy optimization step in RLHF and instead directly expresses the reward modeling step in terms of $\pi_\theta(\boldsymbol{x})$. The authors point out that the KL-controlled policy optimization stage

$$\theta \leftarrow \max_\theta \quad \mathbb{E}_{\pi_\theta(\boldsymbol{x})}\Big[r_\phi(\boldsymbol{x})\Big] - \beta D_{\text{KL}}\big(\pi_\theta(\boldsymbol{x}) \,||\, \pi_{\text{pre}}(\boldsymbol{x})\big) \tag{13}$$

has a closed-form solution

$$\pi_\theta(\boldsymbol{x}) \propto \pi_{\text{pre}}(\boldsymbol{x}) \exp\left(\frac{1}{\beta} r_\phi(\boldsymbol{x})\right). \tag{14}$$

Solving for $r_\phi(\boldsymbol{x})$ and performing the substitution in Equation 1, they obtain a loss function for the reward which expresses the probability that $\boldsymbol{x}_A \succ \boldsymbol{x}_B$ as

$$p_\theta(y; \boldsymbol{x}_A, \boldsymbol{x}_B) = \sigma\left(\beta \log \frac{\pi_\theta(\boldsymbol{x}_A)}{\pi_{\text{pre}}(\boldsymbol{x}_A)} - \beta \log \frac{\pi_\theta(\boldsymbol{x}_B)}{\pi_{\text{pre}}(\boldsymbol{x}_B)}\right). \tag{15}$$

This equation effectively defines a generative process for pairwise preferences for $\pi_\theta(\boldsymbol{x})$ that is guided by a PBDE with $f(\boldsymbol{x}) = \beta$ and $g(\boldsymbol{x}) = -\beta \log \pi_{\text{pre}}(\boldsymbol{x})$, i.e.,

$$\Omega_{\pi_\theta}(\boldsymbol{x}) = \beta \log \pi_\theta(\boldsymbol{x}) - \beta \log \pi_{\text{pre}}(\boldsymbol{x}). \tag{16}$$

If we assume the preference generative process for the annotator is governed by the same PBDE—which could be the case if annotator preferences express only *relative* improvements that correct for the base fluency of the outputs—then Theorem 2 holds and the model recovers $\pi_\theta(\boldsymbol{x}) = p^*(\boldsymbol{x})$ (Figure 2).

Note that different assumptions about the preference generative processes can lead to a different relationship between $p^*$ and $\pi_\theta$, even if those assumptions produce the same observed preference distribution $p(y; \boldsymbol{x}_A, \boldsymbol{x}_B)$. For instance, if we instead assume the annotator behaves according to the Luce choice rule, but still minimize the DPO objective, the globally optimal $\pi_\theta(\boldsymbol{x})$ would become

$$\pi_\theta(\boldsymbol{x}) \propto \pi_{\text{pre}}(\boldsymbol{x}) \cdot p^*(\boldsymbol{x})^{\frac{1}{\beta}}. \tag{17}$$

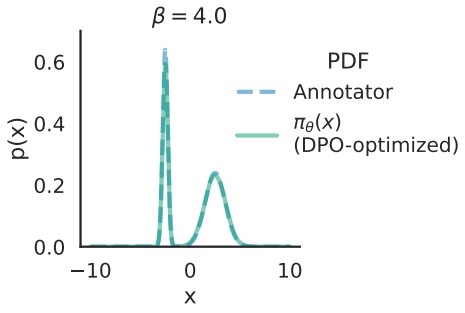

Figure 2: Theorem 2 also holds for DPO if the annotator and model share the same generative process (Equation 15).

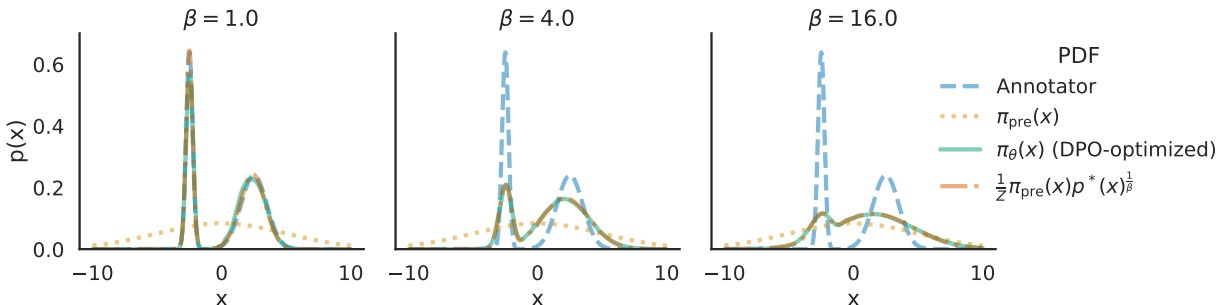

Figure 3: Under the Luce choice rule assumption for the annotator's generative process on pairwise prefer­ences, using DPO to tune a generative model $\pi_{\mathrm{pre}}(\boldsymbol{x})$ (dotted orange) on preferences derived from the implicit preference distribution (dashed blue) results in a mixture of experts model between the initial model $\pi_{\mathrm{pre}}$ and the temperature-smoothed implicit preference distribution, as demonstrated by the agreement between the empirical (solid green) and theoretical (dash-dotted red) curves.

A proof is provided in Appendix A.1. Under this assumption, DPO converges to a product of experts between a pretrained "prior" policy $\pi_{\mathrm{pre}}(\boldsymbol{x})$ and the annotator's temperature-smoothed implicit preference distribution $p^*(\boldsymbol{x})$. This means that whether or not $\pi_\theta(\boldsymbol{x})$ converges to $p^*(\boldsymbol{x})$ depends as much on our assumptions about the preference generating process as it does on the algorithm we use.

We illustrate this by finetuning a pretrained model $\pi_{\mathrm{pre}}(\boldsymbol{x})$ with DPO on pairwise preferences acquired from a synthetic annotator whose generative process for pairwise preferences obeys the Luce choice rule with an implicit preference distribution defined in Equation 10 (Figure 3, dashed blue). The model $\pi_{\mathrm{pre}}(\boldsymbol{x})$ (Figure 3, dotted orange) is an energy-based model with an energy function expressed as a four-layer MLP (we reuse hyperparameters detailed in Table 1):

$$\pi_{\mathrm{pre}}(\boldsymbol{x}) = \frac{1}{Z_{\mathrm{pre}}} e^{-\mathrm{MLP}_{\mathrm{pre}}(\boldsymbol{x})}. \tag{18}$$

We initialize it by regressing on the log-likelihood of $N_{\mathrm{truncated}}(\mu = 0.0, \sigma = 5.0)$ defined over $[-10, 10]$, and we sweep over $\beta \in \{1, 4, 16\}$ to adapt it with DPO. The adapted model (Figure 3, solid green) behaves just as predicted by Equation 17 (Figure 3, dash-dotted orange).

We point out that Equation 17 is almost a weighted geometric mean but is missing a $1 - \frac{1}{\beta}$ exponent on $\pi_{\mathrm{pre}}(\boldsymbol{x})$. If we were instead to define $f(\boldsymbol{x}) = \alpha^{-1}$ and $g(\boldsymbol{x}) = (1 - \alpha^{-1}) \log \pi_{\mathrm{pre}}(\boldsymbol{x})$, the PBDE for the model would result in a globally optimal $\pi_\theta(\boldsymbol{x})$ (again, assuming the annotator behaves according to the Luce choice rule) of the form

$$\pi_\theta(\boldsymbol{x}) \propto \pi_{\mathrm{pre}}(\boldsymbol{x})^{1-\alpha} \cdot p^*(\boldsymbol{x})^\alpha. \tag{19}$$

Refer to Appendix A.1 for a proof and Figure 8 in the Appendix for an empirical learning behavior visual­ization in that case.

## 5 Annotator misspecification

In Section 4.3 we touched on cases where the mismatch generative processes for pairwise preferences between the annotator and the model was used to our advantage to induce specific properties in the globally optimal model. In this section, we show that an unwanted mismatch in generative processes (which we call *annotator misspecification*) can lead to badly tuned policies.

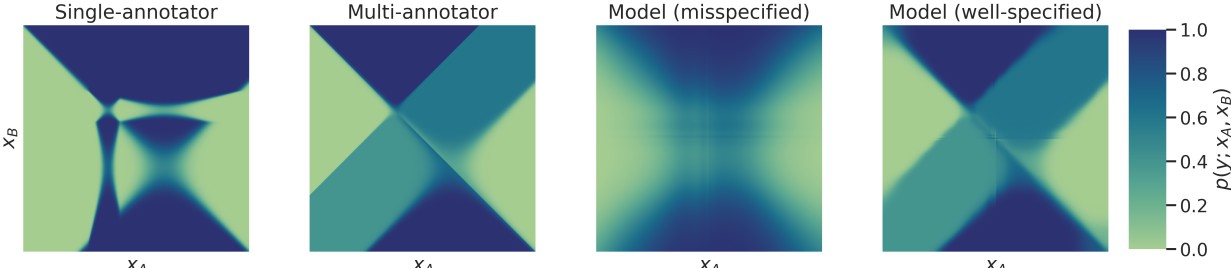

Figure 4: $\text{Prob}(\boldsymbol{x}_A \succ \boldsymbol{x}_B)$ for single-annotator (Equation 20) and multi-annotator (Equation 21) behavior (left two plots) as well as models adapted with a misspecified and well-specified annotator behavior model (right two plots). The large regions of near-0.5 probability in the multi-annotator case (second plot) are caused by strong but opposing preferences for the two annotators, which cannot be captured by a single-annotator reward model (third plot).

## 5.1 Annotator misspecification in a toy setting

Consider a scenario where two annotators—Annotator 1 and Annotator 2—are providing feedback on coffees served at various temperatures. Annotator 1 happens to really like iced coffee, while Annotator 2 prefers hot coffee. In the experiment, an annotator is selected at random and is served a pair of coffees at temperatures sampled independently at random. The annotator expresses a preference which is added to the feedback dataset alongside the two coffee temperatures, but *without noting the annotator's identity*.

We simulate this by modifying the way in which feedback is acquired from Equation 10: instead of having a single annotator following the Luce choice rule with an implicit preference distribution which covers two modes, we assume the distribution of *preferences* is itself a mixture of two preference distributions, each of which is obtained from a distinct annotator following the Luce choice rule with an implicit preference distribution covering one of the two modes. Mathematically speaking, instead of drawing preferences from

$$\text{Prob}(\boldsymbol{x}_A \succ \boldsymbol{x}_B) = \frac{p^*(\boldsymbol{x}_A)}{p^*(\boldsymbol{x}_A) + p^*(\boldsymbol{x}_B)} = \frac{\sum_{i=1}^{2} w_i \cdot p_i^*(\boldsymbol{x}_A)}{\sum_{i=1}^{2} w_i \cdot p_i^*(\boldsymbol{x}_A) + \sum_{i=1}^{2} w_i \cdot p_i^*(\boldsymbol{x}_B)}, \tag{20}$$

(where $w_1 = 2/5$, $p_1^* = N_{\text{truncated}}(\mu = -2.5, \sigma = 0.25)$, $w_2 = 3/5$, and $p_2^* = N_{\text{truncated}}(\mu = 2.5, \sigma = 1.0)$), we now sample preferences from a mixture of two separate preference distributions

$$\text{Prob}(\boldsymbol{x}_A \succ \boldsymbol{x}_B) = \sum_{i=1}^{2} w_i \cdot \text{Prob}_i(\boldsymbol{x}_A \succ \boldsymbol{x}_B) = \sum_{i=1}^{2} w_i \cdot \frac{p_i^*(\boldsymbol{x}_A)}{p_i^*(\boldsymbol{x}_A) + p_i^*(\boldsymbol{x}_B)} \tag{21}$$

A visual way to reason about the differences between Equations 20 and 21 is to plot $\text{Prob}(\boldsymbol{x}_A \succ \boldsymbol{x}_B)$ in both cases (Figure 4, left half). A cursory examination shows that the two result in very different comparison outcome statistics.

If we adapt $\pi_{\text{pre}}(\boldsymbol{x})$ on comparison outcomes sampled from Equation 21 using a Luce choice rule generative process for the model, we are in a annotator misspecification setting and the adapted model (Figure 5, solid green) fails to capture regions of high density under either of the two annotators' PDFs (Figure 5, dashed blue and dot-dashed pink). In other words, we learn to serve lukewarm coffee to both annotators despite neither of them enjoying it. This is also reflected in the

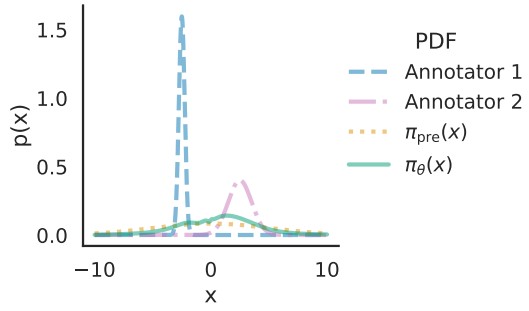

Figure 5: If preferences are aggregated across two annotators (dashed blue, dot-dashed pink) but the model is adapted under a single-annotator assumption, it fails to capture either annotator's PDF (solid green).

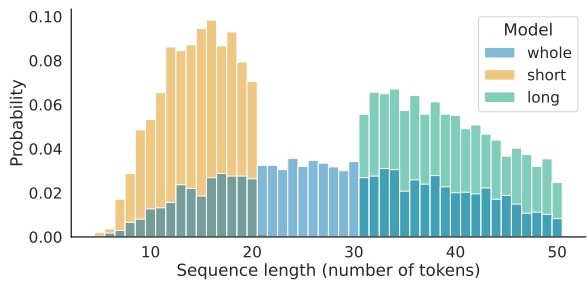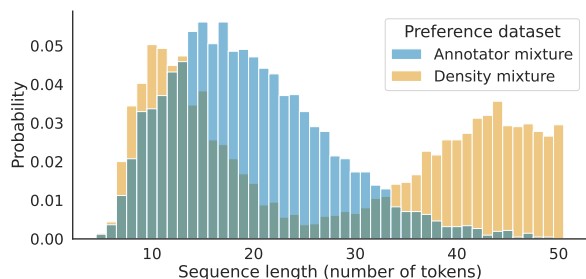

Figure 6: Sentence length distributions. (Left) Histogram of sequence lengths for sequences sampled from the *whole*, *short*, and *long* model checkpoints. (Right) Histogram of sequence lengths after adapting the pretrained model $\pi_{\mathrm{pre}}(\boldsymbol{x})$ (the *whole* model checkpoint) on two distinct preference datasets: the "annotator mixture" dataset (Equation 22) and the "density mixture" dataset (Equation 23).

comparison outcome statistics captured by the model (Figure 4, third plot). Addressing annotator misspecification is an open problem, especially in the absence of annotator IDs. In Appendix A.3 we present results where we succeed in disentangling the two annotators' preferences without annotator IDs (also refer to Figure 4's rightmost plot), but applying the same approach proved unsuccesful in the larger-scale setting which we will discuss shortly.

This failure mode of learning from pairwise preferences is particularly relevant in a context where LLMs are aligned with human preferences by crowdsourcing the acquisition of pairwise preferences across multiple human annotators. It could appear at first glance that the resulting preferences are inconsistent with one another when in fact annotator identity is a confounding factor that needs to be taken into account. For instance, consider a setting where a group of human annotators are asked to rank pairs of model outputs in terms of fluency. Given equally fluent outputs, some annotators may be biased towards short sentences while others may be biased towards longer sentences.

## 5.2 Annotator misspecification in a language modeling setting

We now move to the One Billion Words Benchmark (LM1B; Chelba et al., 2013) to illustrate the consequences of annotator misspecification in a language modeling setting. Starting from Flax's LM1B example code[6] we train three separate instances of a Transformer-based architecture on distinct subsets of LM1B according to sequence length: less than 50 tokens (*whole*), between 1 and 20 tokens inclusively (*short*), and between 30 and 50 tokens inclusively (*long*). We treat the *whole* model as the pretrained LM to adapt via pairwise preference feedback and use the *short* and *long* models to form synthetic annotators. Figure 6 (left) shows sequence length distributions for each model.

Recall that for variable-length inputs the Luce choice rule results in "length bias" (in that applying it to a policy leads to "length bias" in the resulting preferences), since longer sequences naturally have lower likelihood magnitudes (subsection 4.3). We illustrate this by comparing sequence length distributions (Figure 6, left) to the probability given by the Luce choice rule (Equation 9) that a synthetic annotator with a *long* implicit preference distribution will prefer short sequences over long ones (Table 2): that probability is around 97%. When comparing the outcome probabilities for short, medium, and long sequences (Table 2), Equation 12 better aligns with the intuitions built from examining Figure 6 (left).

We now revisit annotator misspecification when pairwise preferences are acquired from multiple annotators, assuming annotators *and* the model behave according to the length-normalized Luce choice rule. We sample $2^{15}$ pairs from the *whole* model and compare the outcome of learning on two preference datasets:

---

[6] https://github.com/google/flax/tree/main/examples/lm1b

Table 2: Probability of a synthetic annotator preferring some token sequence over another token sequence as a function of their respective lengths. Token sequences are sampled from the pretrained LM (*whole*) and are binned into small (S, 20 tokens or less), medium (M, between 21 and 29 tokens), and large (L, 30 tokens or more) sets of sequences. $P(X \succ Y)$ represents an average of $P(x_A \succ x_B)$ over all $x_A \in X, x_B \in Y$.

| Synthetic Annotator | $P(S \succ M)$ | | $P(S \succ L)$ | | $P(M \succ L)$ | |
| | Eqn 9 | Eqn 12 | Eqn 9 | Eqn 12 | Eqn 9 | Eqn 12 |
|---|---|---|---|---|---|---|
| *whole* | 0.9094 | 0.4042 | 0.9928 | 0.3621 | 0.9175 | 0.4554 |
| *short* | 0.9906 | 0.9540 | 1.0000 | 1.0000 | 0.9945 | 0.9783 |
| *long* | 0.9190 | 0.2489 | 0.9729 | 0.0533 | 0.7873 | 0.1575 |

- The "annotator mixture" dataset is built as a uniform mixture of preferences sampled independently using the *short* and *long* model checkpoints. For each $(\boldsymbol{x}_A, \boldsymbol{x}_B)$ pair, we first sample uniformly at random which checkpoint will be used as the implicit preference distribution, and we then sample a preference using the length-normalized Luce choice rule:

$$p(y; \boldsymbol{x}_A, \boldsymbol{x}_B) = \sum_{\theta_{\text{pre}} \in \{\theta_{\text{short}}, \theta_{\text{long}}\}} \frac{1}{2} \cdot \sigma \left( \frac{\log \pi_{\theta_{\text{pre}}}(\boldsymbol{x}_A)}{|\boldsymbol{x}_A|} - \frac{\log \pi_{\theta_{\text{pre}}}(\boldsymbol{x}_B)}{|\boldsymbol{x}_B|} \right). \tag{22}$$

- The "density mixture" dataset is built following the length-normalized Luce choice rule with an implicit preference distribution that is a uniform mixture of the distributions computed by the *short* and *long* model checkpoints:

$$p(y; \boldsymbol{x}_A, \boldsymbol{x}_B) = \sigma \left( \frac{\log \pi_{\text{mix}}(\boldsymbol{x}_A)}{|\boldsymbol{x}_A|} - \frac{\log \pi_{\text{mix}}(\boldsymbol{x}_B)}{|\boldsymbol{x}_B|} \right), \quad \pi_{\text{mix}}(\boldsymbol{x}) = \frac{\pi_{\theta_{\text{short}}}(\boldsymbol{x}) + \pi_{\theta_{\text{long}}}(\boldsymbol{x})}{2}. \tag{23}$$

In other words, the "annotator mixture" dataset contains a mixture of preferences from a synthetic annotator who likes short sequences and a synthetic annotator who likes long sequences, whereas the "density mixture" dataset contains preferences from a single synthetic annotator who likes both short and long sequences, but not medium-length sequences. Figure 6 (right) shows sequence length distributions after adapting on either of the two preference datasets. We notice that the model adapted on the "annotator mixture" preference dataset generates medium-length sentences much more frequently than either the *short* or *long* model checkpoints, exactly like in the "cold and hot coffee" toy example. This is a direct result of annotator misspecification: pairwise preferences are obtained by querying two synthetic annotators, whereas the learning rule used for adaptation assumes a generative process for preferences that involves a *single* annotator. On the other hand, when learning from preferences acquired from a synthetic annotator whose implicit preference distribution is a mixture of the *short* and *long* model checkpoints ("density mixture" dataset), the generative processes for pairwise preferences are the same across annotator and model, and the model succeeds in learning to output a bimodal distribution of sequence lengths.

## 6 Discussion and Limitations

In this work we use a simplified, synthetic experimental framework to empirically verify the theoretical properties of approaches which learn from pairwise human feedback. Specifically, we

1. define a probability distribution over a random variable with finite support and use it as the implicit preference distribution for a synthetic annotator following a fixed preference generating process;

2. provide the model with access to the implicit preference distribution only through a fixed number of queries to the synthetic annotator on observation pairs which are ranked stochastically using a PBDE; and

3. adapt the model on the resulting $(\boldsymbol{x}_A, \boldsymbol{x}_B, y)$ triplets.

In this synthetic setting, we have shown that it is indeed possible to approximately recover the annotator's implicit preference distribution. We have also demonstrated how this procedure fails in the presence of annotator misspecification.

Solving annotator misspecification in the presence of multiple annotators requires unsupervised discovery of annotator clusters and is a difficult and open problem. The naive approach we present in Appendix A.3 requires knowing the correct number of clusters in advance and represents an important drawback. The literature on probabilistic modeling suggests that we could perhaps replace the categorical random variable random for the cluster ID with a continuous random variable, but this has yet to be tried in practice.

In some cases practitioners could conceivably use additional metadata to their advantage, for instance by assigning each annotator with a unique ID and solving the binary preference classification problem conditioned on the annotator ID. Alternatively, if practitioners know in advance that preferences are polarized and if the way each annotator leans can be predicted from certain covariates (age, location, etc.), those covariates could be used as additional context in the binary preference classification problem.

Determining whether behavioral assumptions are correct is also a difficult problem. Practitioners' decisions should be informed by results in disciplines like mathematical psychology or by their empirical knowledge. Additionally, should a model fail to learn from pairwise preferences, the possibility that a practitioner's behavioral assumptions are wrong should at least be considered.

Beyond annotator misspecification, there are a number of other ways our synthetic density estimation setting differs from practical applications of learning from pairwise human feedback.

**Finite data**   We do not account for the labor necessary to acquire large amounts of human preference data. The theoretical results we present hold in the infinite data limit and the existence of a global minimizer for the loss function does not guarantee convergence through gradient descent. The univariate toy experiment (Figure 1) is carried at the larger end of the data regime spectrum given the dimensionality of the problem and the number of queries made to the synthetic annotator, and in that setting we achieve convergence.

Practical applications are high-dimensional (images, language tokens, etc.), and the number of synthetic annotator queries we make in our LM1B experiment is comparatively small: the size of the vocabulary ($30\,000$) and the maximum sequence length ($50$) result in a set of possible sequences that is considerably larger than the number of synthetic annotator queries ($2^{15}$). It is interesting to note, then, that our LM1B experiment (and more broadly contemporary works on adapting LLMs to pairwise human feedback) are reasonably effective at steering models from pairwise feedback alone. How this is possible despite the combinatorial nature of language is an interesting question for future work. Plausible hypotheses include:

- Natural language conforms to the manifold hypothesis (Cayton, 2005; Narayanan & Mitter, 2010), and the pretrained model $\pi_{\mathrm{pre}}(\boldsymbol{x})$ is a particularly good proposal distribution to explore the "manifold" of natural language. The result is that characterizing a distribution over natural language outputs requires far fewer pairwise comparisons than the curse of dimensionality would suggest.

- Density estimation using pairwise human feedback *is* really hard in the data regimes in which it is deployed, but it is not *necessary* to fully capture the implicit preference distribution in order to steer the generative model in the right direction. Provided careful regularization is applied (which, to our knowledge, is a requirement in practice), preference datasets are informative enough.

**Stationarity**   Our experiments assume all observation pairs are sampled at once before adapting the model. In practice some approaches (like RLHF) use a nonstationary proposal distribution and alternate between querying for human preferences using sample pairs drawn from the model and adapting the model on human preferences. This raises interesting questions on whether stationarity is preferable, and whether active learning could play a role in reducing the amount of annotator queries necessary for good performance.

# 7 Conclusion

In this work we present a probabilistic modeling as an alternative to reinforcement learning for learning from pairwise human preferences. For a broad family of generative processes guided by what we call preference behavior distribution equations (PBDEs) we show through theoretical and empirical results that, so long as the human annotator and the model share the same PBDE for generating pairwise preferences, the global optimum of the resulting optimization problem is the annotator's implicit preference distribution over observations. We argue that being explicit about the assumed generative process for an annotator's preferences is important both to manipulate the properties of the globally optimal model that learns on those preferences *and* to avoid annotator misspecification failure modes. Finally, we illustrate the practical implications of annotator misspecification in a language modeling problem where the usual single-annotator assumption fails to adequately capture two annotators' conflicting preferences.

## Broader Impact Statement

Our work investigates learning from pairwise human preferences from a theoretical standpoint, but it has multiple practical implications. Density estimation from pairwise feedback is a very difficult problem—arguably much more so than density estimation from target distribution samples—and most practical applications operate in the low-data regime. This has the potential to exacerbate biases in the pairwise preference data and makes it all the more important to ensure that annotators are representative of a diverse population of end-users. The insights we provide on annotator misspecification also highlight the importance of reasoning explicitly about the generative process for pairwise preferences, as a misspecified process has the potential to steer the model towards unwanted behavior or erase underrepresented perspectives altogether. We believe future research on both these aspects of learning from pairwise human preferences is important in order to achieve inclusive alignment with human judgement.

## Author Contributions

**Vincent Dumoulin** wrote a first draft of the manuscript which co-authors then added to and helped edit. He implemented and carried out the experiments (starting from Yann's code and model checkpoints in the case of LM1B). He derived the proofs for Theorem 1 (with input from Laurent Dinh) and Theorem 2 (with input from Daniel) and developed the length-normalized Luce choice rule solution to the "length bias" problem jointly with Yann.

**Daniel D. Johnson** identified the problem of annotator misspecification with two annotators and proposed the "family of generative processes" framing to unify the Luce choice rule, DPO, and the length-normalized Luce choice rule. He contributed to writing and editing, and helped with some of the proofs. He explored alternative variants of the LM1B experiment which ended up not making it into the paper.

**Pablo Samuel Castro** provided regular input in the research discussions and helped write and edit the manuscript.

**Hugo Larochelle** proposed the research direction that eventually led to the paper's narrative of framing learning from pairwise human preferences as a probabilistic modeling problem. He advised on the project and helped revise the manuscript.

**Yann Dauphin** helped with experiments and trained the LM1B model checkpoints used in the submission. He proposed latent variables as a solution to user misspecification with two annotators (Section A.3). He identified the "length bias" issue with the Luce choice rule for autoregressive models and he developed the length-normalized Luce choice rule solution jointly with Vincent. He helped develop and frame the theoretical results in the paper.

## Acknowledgements

We would like to thank Laurent Dinh for his help in deriving the proof for Theorem 1; Jesse Engel for his input in the early stages of the project; Christian Walder for his input on inducing specific properties in globally optimal models by manipulating the model's generative process for preferences; Ben Poole for his

thoughtful feedback in revising the manuscript; and Rishabh Joshi for clarifying the implementation details of SLiC-HF-direct.

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

# A Appendix

## A.1 Theorems and proofs

**Lemma 1.** *Let $q(\boldsymbol{x}_A, \boldsymbol{x}_B)$ be a joint probability distribution with support $\mathcal{S} \times \mathcal{S}$ and for which $q(\boldsymbol{x}_A, \boldsymbol{x}_B) > 0$ for all $\boldsymbol{x}_A, \boldsymbol{x}_B \in \mathcal{S} \times \mathcal{S}$. Let $\chi(\boldsymbol{x}) \colon \mathcal{S} \to \mathbb{R}$ and $\omega(\boldsymbol{x}) \colon \mathcal{S} \to \mathbb{R}$ be two injective functions mapping $\boldsymbol{x}$ to scalar "scores". Let*

$$p_\chi(y; \boldsymbol{x}_A, \boldsymbol{x}_B) = \frac{e^{\chi(\boldsymbol{x}_A)}}{e^{\chi(\boldsymbol{x}_A)} + e^{\chi(\boldsymbol{x}_B)}} = \sigma\Big(\chi(\boldsymbol{x}_A) - \chi(\boldsymbol{x}_B)\Big) \quad \text{and} \quad p_\omega(y; \boldsymbol{x}_A, \boldsymbol{x}_B) = \sigma\Big(\omega(\boldsymbol{x}_A) - \omega(\boldsymbol{x}_B)\Big).$$

*The loss function*

$$\mathbb{E}_{\boldsymbol{x}_A, \boldsymbol{x}_B \sim q(\boldsymbol{x}_A, \boldsymbol{x}_B)}\Big[D_{\mathrm{KL}}\big(p_\chi(y; \boldsymbol{x}_A, \boldsymbol{x}_B) \,\|\, p_\omega(y; \boldsymbol{x}_A, \boldsymbol{x}_B)\big)\Big].$$

*is globally minimized when*

$$\chi(\boldsymbol{x}) = \omega(\boldsymbol{x}) + C, \quad C \in \mathbb{R}, \forall \boldsymbol{x} \in \mathcal{S}.$$

*Proof.* We know from the properties of the KL-divergence that it is minimized when the left- and right-hand side represent the same distribution, which means that the expectation itself has a global minimum at zero when

$$
\begin{aligned}
\frac{e^{\chi(\boldsymbol{x}_A)}}{e^{\chi(\boldsymbol{x}_A)} + e^{\chi(\boldsymbol{x}_B)}} &= \frac{e^{\omega(\boldsymbol{x}_A)}}{e^{\omega(\boldsymbol{x}_A)} + e^{\omega(\boldsymbol{x}_B)}}, & \forall \boldsymbol{x}_A, \boldsymbol{x}_B \in \mathcal{S} \times \mathcal{S} \\
\Rightarrow e^{\chi(\boldsymbol{x}_A) + \omega(\boldsymbol{x}_A)} + e^{\chi(\boldsymbol{x}_A) + \omega(\boldsymbol{x}_B)} &= e^{\chi(\boldsymbol{x}_A) + \omega(\boldsymbol{x}_A)} + e^{\chi(\boldsymbol{x}_B) + \omega(\boldsymbol{x}_A)}, & \forall \boldsymbol{x}_A, \boldsymbol{x}_B \in \mathcal{S} \times \mathcal{S} \\
\Rightarrow \chi(\boldsymbol{x}_A) &= \omega(\boldsymbol{x}_A) + (\chi(\boldsymbol{x}_B) - \omega(\boldsymbol{x}_B)), & \forall \boldsymbol{x}_A, \boldsymbol{x}_B \in \mathcal{S} \times \mathcal{S}
\end{aligned}
$$

For any given $\boldsymbol{x}_A$, the above equation needs to hold for all $\boldsymbol{x}_B \in \mathcal{S}$, which is only possible if the term $\chi(\boldsymbol{x}_B) - \omega(\boldsymbol{x}_B)$ is constant for all $\boldsymbol{x}_B \in \mathcal{S}$. We therefore conclude that

$$\chi(\boldsymbol{x}) = \omega(\boldsymbol{x}) + C, \quad C \in \mathbb{R}, \forall \boldsymbol{x} \in \mathcal{S}.$$

$\square$

**Proof for Theorem 1**

*Proof.* The proof follows directly from applying Lemma 1 with $\chi(\boldsymbol{x}) = \log p^*(\boldsymbol{x})$ and $\omega(\boldsymbol{x}) = r_\phi(\boldsymbol{x})$: at the global optimum we have that

$$\log p^*(\boldsymbol{x}) = r_\phi(\boldsymbol{x}) + C, \quad C \in \mathbb{R}, \forall \boldsymbol{x} \in \mathcal{S},$$

which means that

$$e^{r_\phi(\boldsymbol{x})} \propto p^*(\boldsymbol{x}) \quad \forall \boldsymbol{x} \in \mathcal{S}.$$

$\square$

**Proof for Theorem 2**

*Proof.* Using Lemma 1 with $\chi(\boldsymbol{x}) = \Omega_{p^*}(\boldsymbol{x})$ and $\omega(\boldsymbol{x}) = \Omega_{\pi_\theta}(\boldsymbol{x})$, at the global optimum we have that

$$
\begin{aligned}
\Omega_{p^*}(\boldsymbol{x}) &= \Omega_{\pi_\theta}(\boldsymbol{x}) + C, & C \in \mathbb{R}, \forall \boldsymbol{x} \in \mathcal{S} \\
\Rightarrow f(\boldsymbol{x}) \log p^*(\boldsymbol{x}) + g(\boldsymbol{x}) &= f(\boldsymbol{x}) \log \pi_\theta(\boldsymbol{x}) + g(\boldsymbol{x}) + C, & C \in \mathbb{R}, \forall \boldsymbol{x} \in \mathcal{S} \\
\Rightarrow p^*(\boldsymbol{x}) &= \pi_\theta(\boldsymbol{x}) e^{\frac{C}{f(\boldsymbol{x})}}, & C \in \mathbb{R}, \forall \boldsymbol{x} \in \mathcal{S}
\end{aligned}
$$

Since both $p^*(\boldsymbol{x})$ and $\pi_\theta(\boldsymbol{x})$ are probability distributions, summing or integrating both sides results in

$$
\mathbb{E}_{\pi_\theta(\boldsymbol{x})} \left[ e^{\frac{C}{f(\boldsymbol{x})}} \right] = 1.
$$

There are three possibilities for $C$: either $C < 0$, $C > 0$, or $C = 0$. Since we know $f(\boldsymbol{x}) > 0$ for all $\boldsymbol{x}$, $C < 0$ would result in an expectation smaller than 1 and $C > 0$ would result in an expectation larger than 1. We conclude that $C = 0$ and therefore

$$
\pi_\theta(\boldsymbol{x}) = p^*(\boldsymbol{x}), \quad \forall \boldsymbol{x}.
$$

$\square$

**Proof of Equation 17's global optimality**

*Proof.* We observe that

$$
p_\theta(y; \boldsymbol{x}_A, \boldsymbol{x}_B) = \sigma\left( \beta \log \frac{\pi_\theta(\boldsymbol{x}_A)}{\pi_{\text{pre}}(\boldsymbol{x}_A)} - \beta \log \frac{\pi_\theta(\boldsymbol{x}_B)}{\pi_{\text{pre}}(\boldsymbol{x}_B)} \right).
$$

Using Lemma 1 with $\chi(\boldsymbol{x}) = \log p^*(\boldsymbol{x})$ and $\omega(\boldsymbol{x}) = \beta(\log \pi_\theta(\boldsymbol{x}) - \log \pi_{\text{pre}}(\boldsymbol{x}))$, at the global optimum we have that

$$
\log p^*(\boldsymbol{x}) = \beta(\log \pi_\theta(\boldsymbol{x}) - \log \pi_{\text{pre}}(\boldsymbol{x})) + C, \quad C \in \mathbb{R}, \forall \boldsymbol{x} \in \mathcal{S},
$$

from which we conclude that

$$
\pi_\theta(\boldsymbol{x}) \propto \pi_{\text{pre}}(\boldsymbol{x}) \cdot p^*(\boldsymbol{x})^{\frac{1}{\beta}}.
$$

$\square$

**Proof of Equation 19's global optimality**

*Proof.* Using Lemma 1 with $\chi(\boldsymbol{x}) = \log p^*(\boldsymbol{x})$ and $\omega(\boldsymbol{x}) = \frac{1}{\alpha}(\log \pi_\theta(\boldsymbol{x}) - \log \pi_{\text{pre}}(\boldsymbol{x})) + \log \pi_{\text{pre}}(\boldsymbol{x})$, at the global optimum we have that

$$
\log p^*(\boldsymbol{x}) = \frac{1}{\alpha}(\log \pi_\theta(\boldsymbol{x}) - \log \pi_{\text{pre}}(\boldsymbol{x})) + \log \pi_{\text{pre}}(\boldsymbol{x}) + C, \quad C \in \mathbb{R}, \forall \boldsymbol{x} \in \mathcal{S},
$$

from which we conclude that

$$
\pi_\theta(\boldsymbol{x}) \propto \pi_{\text{pre}}(\boldsymbol{x})^{1-\alpha} \cdot p^*(\boldsymbol{x})^\alpha.
$$

$\square$

## A.2   Alternatives to the binary cross-entropy loss

Other recent works adapt the model directly on pairwise preferences but move beyond a binary cross-entropy loss formulation. As such, Theorem 2 is not applicable, and more work is needed to characterize their learning properties when viewed through the density estimation lens.

**Rank Responses to align Human Feedback** (RRHF; Yuan et al., 2023) uses a ranking loss instead of the binary cross-entropy loss and considers annotator rankings of $k$ observations $\boldsymbol{x}_1$ through $\boldsymbol{x}_k$:

$$\mathcal{L}_{\text{rank}}(\theta) = \mathbb{E}_{D(\boldsymbol{x}_1,\ldots,\boldsymbol{x}_k)}\left[\sum_{\boldsymbol{x}_i \succ \boldsymbol{x}_j} -\left(\frac{1}{|\boldsymbol{x}_i|}\log\pi_\theta(\boldsymbol{x}_i) - \frac{1}{|\boldsymbol{x}_j|}\log\pi_\theta(\boldsymbol{x}_j)\right)^+\right], \tag{24}$$

where $|\boldsymbol{x}|$ is the length of the token sequence $\boldsymbol{x}$. An additional loss component in the form of negative log-likelihood on the highest-ranked $\boldsymbol{x}_i$ is also used.

**Sequence Likelihood Calibration from Human Feedback** (SLiC-HF; Zhao et al., 2023) considers (among other alternatives) a SLiC-HF-direct variant which optimizes $\pi_\theta(\boldsymbol{x})$ directly on pairwise human preferences. The loss function is defined as

$$\mathcal{L}(\theta) = \mathbb{E}_{D(\boldsymbol{x}_A,\boldsymbol{x}_B,y)}\left[\left(\delta - \left(y\cdot\log\frac{\pi_\theta(\boldsymbol{x}_A)}{\pi_\theta(\boldsymbol{x}_B)} + (1-y)\cdot\log\frac{\pi_\theta(\boldsymbol{x}_B)}{\pi_\theta(\boldsymbol{x}_A)}\right)\right)^+\right] - \lambda\cdot\mathbb{E}_{D_{\text{SFT}}}(\boldsymbol{x})\left[\log\pi_\theta(\boldsymbol{x})\right]. \tag{25}$$

The first expectation represents a hinge loss over the difference in log-likelihoods between $\boldsymbol{x}_A$ and $\boldsymbol{x}_B$: it encourages the difference to be positively large in favor of the higher-ranked observation, up to a margin $\delta$. The second expectation regularizes $\pi_\theta(\boldsymbol{x})$ towards a supervised finetuned (SFT) model trained on a dataset $D_{\text{SFT}}$ of target domain examples, either using the supervised finetuning data directly, drawing samples from the supervised finetuned model, or maximizing the likelihood of preferred observations.

Ignoring the regularization term, the SLiC-HF-direct loss function resembles Equation 1 but uses a hinge loss on the difference in log-likelihoods between $\boldsymbol{x}_A$ and $\boldsymbol{x}_B$ rather than the sigmoid binary cross-entropy. We examine the properties of SLiC-HF-direct using our synthetic density estimation problem by finetuning the pretrained model $\pi_{\text{pre}}(\boldsymbol{x})$ with SLiC-HF-direct on pairwise preferences acquired from the implicit preference distribution defined in Equation 10. Since we consider a purely preference-based setting, we cannot assume access to a finetuning dataset, however we can still regularize the model towards the pretrained model by sampling from $\pi_{\text{pre}}(\boldsymbol{x})$ in Equation 25's second expectation. We sweep over $\delta \in \{0.0, 0.25, 0.5, 1.0, 2.0\}$ and $\lambda \in \{0.0, 1.0\}$ to adapt the pretrained model with SLiC-HF-direct. Figure 9 shows the effect of $\delta$ and $\lambda$. The $\delta$ hyperparameter has a non-uniform smoothing effect on $\pi_\theta(\boldsymbol{x})$, as it bounds the maximum difference in log-probabilities allowed between any two observations. The $\lambda$ hyperparameter has the expected effect of regularizing the adapted model towards $\pi_{\text{pre}}(\boldsymbol{x})$.

**Statistical Rejection Sampling Optimization** (RSO; Liu et al., 2023b) samples from the optimal policy using samples from the pretrained model $\pi_{\text{pre}}(\boldsymbol{x})$, but Liu et al. (2023b) also investigate an extension to SLiC-FH which normalizes the model's log-likelihoods using the pretrained model $\pi_{\text{pre}}(\boldsymbol{x})$:

$$\mathcal{L}(\theta) = \mathbb{E}_{D(\boldsymbol{x}_A,\boldsymbol{x}_B,y)}\left[\left(\delta - \left(y\cdot\log\frac{\pi_\theta(\boldsymbol{x}_A)\pi_{\text{pre}}(\boldsymbol{x}_B)}{\pi_\theta(\boldsymbol{x}_B)\pi_{\text{pre}}(\boldsymbol{x}_A)} + (1-y)\cdot\log\frac{\pi_\theta(\boldsymbol{x}_B)\pi_{\text{pre}}(\boldsymbol{x}_A)}{\pi_\theta(\boldsymbol{x}_A)\pi_{\text{pre}}(\boldsymbol{x}_B)}\right)\right)^+\right]. \tag{26}$$

**ΨPO with identity mapping** (IPO; Azar et al., 2023) derives a learning rule for a KL-controlled preference probability maximization problem

$$\theta \leftarrow \max_\theta \quad \mathbb{E}_{\substack{\boldsymbol{x}_A \sim \pi_\theta(\boldsymbol{x}) \\ \boldsymbol{x}_B \sim \mu(\boldsymbol{x})}}\left[p^*(y;\boldsymbol{x}_A,\boldsymbol{x}_B)\right] - \tau D_{\text{KL}}\left(\pi_\theta(\boldsymbol{x}) \,||\, \pi_{\text{pre}}(\boldsymbol{x})\right) \tag{27}$$

which has the form

$$\mathcal{L}(\theta) = \mathbb{E}_{D(\boldsymbol{x}_A, \boldsymbol{x}_B, y)} \left[ \left( y \cdot \log \frac{\pi_\theta(\boldsymbol{x}_A)\pi_{\text{pre}}(\boldsymbol{x}_B)}{\pi_\theta(\boldsymbol{x}_B)\pi_{\text{pre}}(\boldsymbol{x}_A)} + (1-y) \cdot \log \frac{\pi_\theta(\boldsymbol{x}_B)\pi_{\text{pre}}(\boldsymbol{x}_A)}{\pi_\theta(\boldsymbol{x}_A)\pi_{\text{pre}}(\boldsymbol{x}_B)} - \frac{\tau^{-1}}{2} \right)^2 \right]. \tag{28}$$

Practically speaking, the loss is similar to the one investigated in SLiC-FH (and RSO) but is constructed using a two-sided quadratic expression rather than a one-sided hinge expression. We examine the properties of IPO using our synthetic density estimation problem by finetuning the pretrained model $\pi_{\text{pre}}(\boldsymbol{x})$ with IPO on pairwise preferences acquired from the implicit preference distribution defined in Equation 10. We sweep over $\tau \in \{0.1, 0.5, 1.0\}$ to adapt the pretrained model with IPO. Figure 10 shows the regularizing effect of $\tau$ towards the pretrained model's density.

### A.3 Addressing annotator misspecification in a toy setting

To account for annotator identity in the experiment described in Section 5, we initialize two copies of $\pi_{\text{pre}}(\boldsymbol{x})$ (adding Gaussian noise with standard deviation $1 \times 10^{-4}$ to break symmetries) to create a mixture model

$$\pi_\theta(\boldsymbol{x}) = \sum_{i=1}^{2} \frac{w_i}{w_1 + w_2} \pi_\theta(\boldsymbol{x}) \tag{29}$$

and adjust our annotator behavior model to account for Annotator 1 and Annotator 2 so that

$$p_\theta(y; \boldsymbol{x}_A, \boldsymbol{x}_B) = \sum_{i=1}^{2} \frac{w_i}{w_1 + w_2} \cdot \frac{\pi_\theta(\boldsymbol{x}_A)}{\pi_\theta(\boldsymbol{x}_A) + \pi_\theta(\boldsymbol{x}_B)}. \tag{30}$$

We then train as usual with a binary cross-entropy loss on the comparison outcome variable $y$ (Figure 11). Doing so allows to recover the correct marginal implicit preference distribution, even in the absence of annotator IDs in the preference data. This is particularly interesting in contexts where human annotators "cluster" in their preferences. In the coffee example presented in section 5, the two annotators may represent entire subpopulations of annotators.

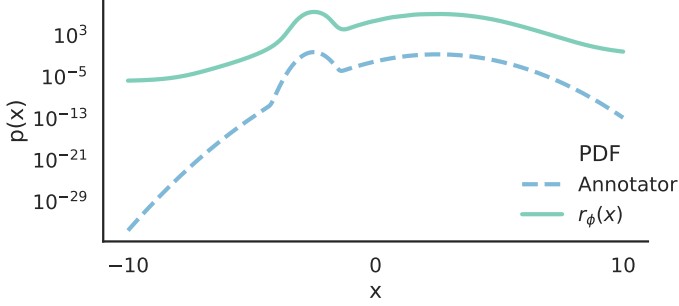

Figure 7: A reward model trained on comparison outcomes sampled according to an implicit preference distribution (Equation 10; dashed blue) converges to the annotator's unnormalized preference distribution (solid green), as indicated by the constant offset between the two curves.

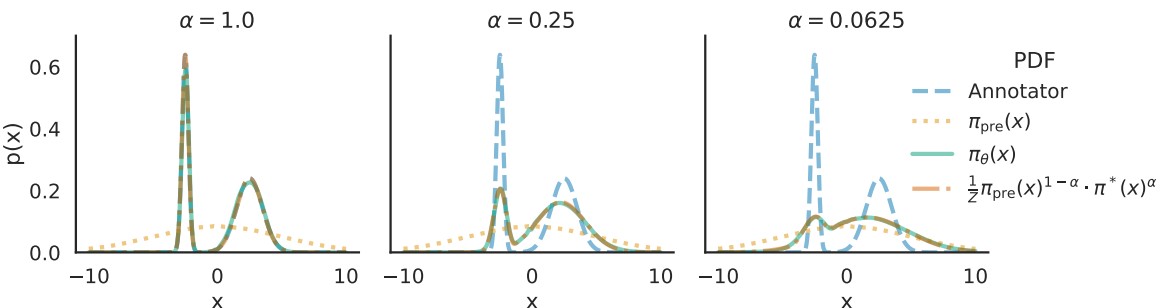

Figure 8: Under the Luce choice rule for the annotator, using a PBDE for the model with $f(\boldsymbol{x}) = \alpha^{-1}$ and $g(\boldsymbol{x}) = (1-\alpha^{-1})\log \pi_{\mathrm{pre}}(\boldsymbol{x})$ to tune $\pi_{\mathrm{pre}}(\boldsymbol{x})$ (dotted orange) on pairwise preferences derived from the implicit preference distribution (dashed blue) results in a weighted geometric average between the initial model $\pi_{\mathrm{pre}}$ and the implicit preference distribution, as demonstrated by the agreement between the empirical (solid green) and theoretical (dash-dotted red) curves.

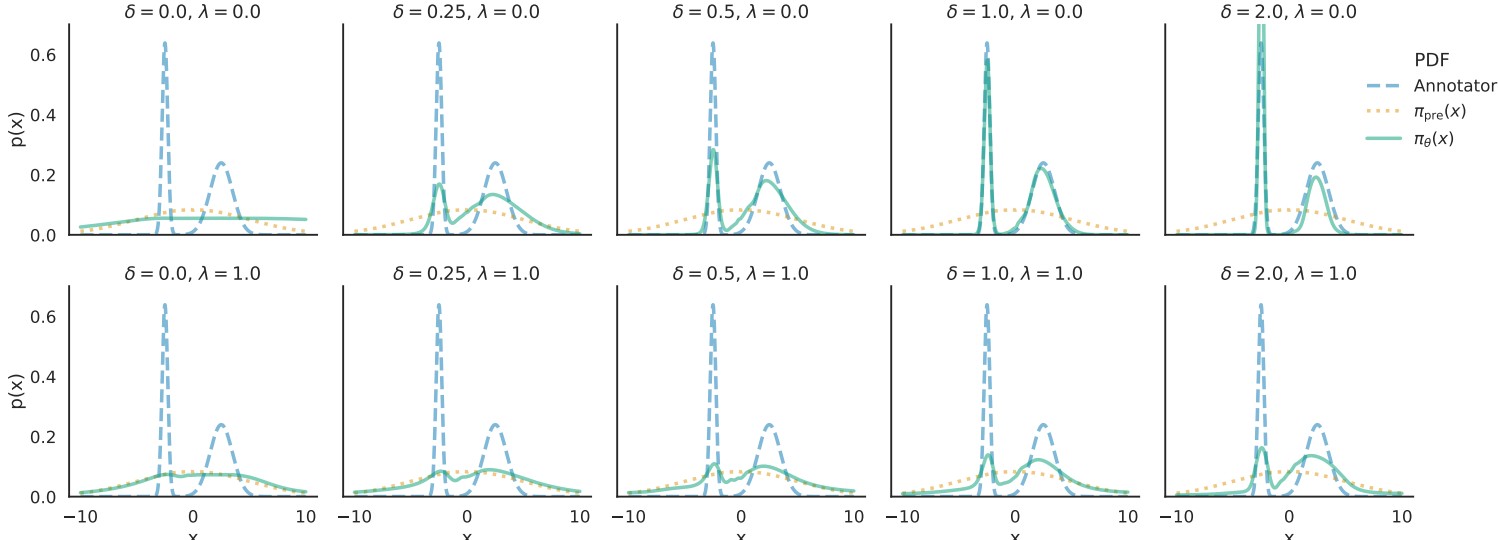

Figure 9: Using SLiC-HF-direct to tune a generative model $\pi_{\mathrm{pre}}(\boldsymbol{x})$ (dotted orange) on pairwise preferences derived from the implicit preference distribution (dashed blue) results in different adapted models (solid green) depending on the choice of hyperparameters $\delta$ and $\lambda$.

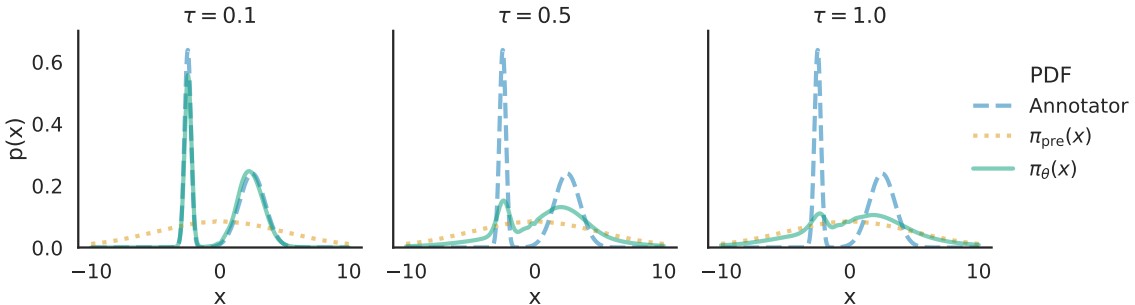

Figure 10: Using IPO to tune a generative model $\pi_{\mathrm{pre}}(\boldsymbol{x})$ (dotted orange) on pairwise preferences derived from the implicit preference distribution (dashed blue) results in different adapted models (solid green) depending on the choice of the $\tau$ hyperparameter.

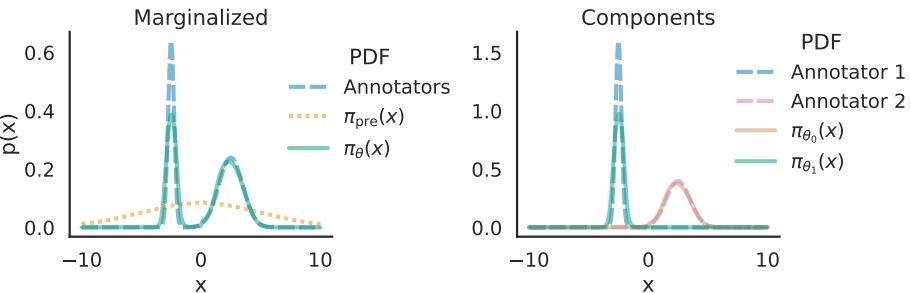

Figure 11: Choosing the correct annotator behavior model (Equation 30) allows to recover the correct marginal implicit preference distribution (left) and individual implicit preference distributions (right), even without annotator IDs in the preference data.

