# OpenReview forum: "A density estimation perspective on learning from pairwise human preferences"
_TMLR — Accepted by TMLR_

### Review · Reviewer_GBeD · 2023-11-28

**Summary Of Contributions:**

This work presents a probabilistic modeling perspective upon the preference behavior distribution equation, which transforms the human feedback into a reward function in the RL framework. I can see that the authors are versed in modeling probabilistic models in proposal distribution and its relevant inference method, as demostrated in Eq 6,7,8. This work will significantly solidify the theoretic basis of the presvious works in Section 2.2.

**Audience:**

Yes

**Broader Impact Concerns:**

no ethical problem

**Claims And Evidence:**

Yes

**Requested Changes:**

I don't need any change at the moment

**Strengths And Weaknesses:**

Strength:

Authors exactly knows their expansion direction as stated in Section 4.3, and it is a classic probabilistic modeling to approach the preference modeling as distributions. They are kind enough to interpret others' methods in their probabilistic framework, i.e. the author's interpretation of Direct Preference Optimization in the density estimation procedure.

Once the preference learning becomes the density estimation problem, all sorts of density estimation tricks can be applied as authors demonstrate in Section 4.4. Sequential likelihood modeling, rejection sampling... All these are classical methods of probabilistic modeling.

I am pretty sure that authors would include a researcher who knows the value of old-fashioned PGM in 2000s and 2010s, which still means a lot in these days, as well.

Good work.

---

> ### Author Response · Authors · 2024-01-10
> **Response from the authors**
>
> Thank you for your encouraging feedback! We remain committed to addressing any additional feedback you may have.

---

### Review · Reviewer_yDZ9 · 2023-12-29

**Summary Of Contributions:**

The paper reviews the modeling assumptions underlying the practice of learning from human feedback in LLM alignment. The authors make the link between assuming that human feedback follows a Luce model and the reward moeling step, and propose a general formula for directly learning the policy from human feedback, which includes as special cases a large number of recent works regarding different loss functions or regularizers.

The authors discuss the effect of "annotator misspecification", meaning that the model of human feedback is incorrect, and in particular show undesirable effects that come from using the usual model in cases where annotators have conflicting preferences. The authors provide an example in which the model turns out to be biased towards shorter outputs.

The paper is moslty theoretical, the experiments are mostly of qualitative nature in artificial settings

**Audience:**

No

**Claims And Evidence:**

Yes

**Requested Changes:**

the paper should go through another round of review anyway

requested changes:
* less space dedicated to the beginning and theorem 1 and 2 since most of it is already known, focus on what's different from DPO/IPO etc.
* the discussion on annotator misspecification is interesting, but without real-life experiments it is unclear what the take away of the paper is

**Strengths And Weaknesses:**

strengths:
* the paper is clear and well-written
* the framework is sound
* the question of annotator misspecification is relevant.

weaknesses:
* the originality and contributions are low
* the experiments are toy
* most importantly, the paper doesn't have much content: theorem 1 and 2 are trivial as far as I can see, the "general formula" is just a general formula, but there is no particular insight into how to choose the specific values, there are two toy experiments with unclear practical value.

Compared to papers coming out lately on similar topics (e.g., papers for DPO, IPO and the likes) this particular paper falls short in technical novelty and experimental insights.

other comments:
--- section 4.1: I don't understand the contribution in that section. This seems to be paraphrasing the fact that the pairwise logistic assumption to learn the reward model is correct under the assumption that human preferences follow the bradley-terry model, this seems to be known and is clearly explained in the DPO paper.

-- Th 1 and 2: I don't understand what is non-trivial in the theorems, it is well-known that  the KL is 0 if and only if the probability distributions are the same

-- 5.1: annotator misspecification: maybe the authors could clarify what result they would expect in cases where there is no agreement between annotators. I also easily believe that in the case of "LLM alignment" where annotators are supposed to follow fairly strict guidelines such extreme disagreement where the middle ground is undesirable doesn't really happen.

-- section 5.2 "the long model assigns higher likelihoods to short token sequences despite
having a near-zero probability of sampling them": this sentence seems inconsistent (high likelihood for sequences of low probability), maybe it's because the authors don't compute an actual likelihood (e.g., without taking into account the probability of "end of sequence" token)?

-- section 5.3: the length bias is fairly obvious and the difference between sum log prob(token|history) vs mean log prob (token|history) is folklore. mean log prob also has its issues (e.g., favoring long sequences with low token-level entropy).

---

> ### Author Response · Authors · 2024-01-10
> **Response from the authors**
>
> Thank you for your constructive feedback!
>
> > the originality and contributions are low / the experiments are toy
>
> The TMLR evaluation criteria state:
>
> "The acceptance decision for a submission is based on the answers to the following questions:
>
> * Are the claims made in the submission supported by accurate, convincing and clear evidence?
> * Would at least some individuals in TMLR's audience be interested in knowing the findings of this paper?
>
> Papers should be accepted if they meet the criteria, _even if the contribution or significance of the work is modest_." (emphasis theirs)
>
> In particular, originality and significance of contributions are not part of the TMLR decision criteria. You do not appear to contest that the claims we make in the submission are supported by accurate, convincing and clear evidence, and you also mention in your review that "the question of annotator misspecification is relevant" and "interesting". We think this is consistent with the standard for an accepted paper at TMLR.
>
> Although our analysis uses several toy experiments, this comment overlooks our experiments in Section 5.2 on language modeling. These experiments involve language modeling on the One Billion Word dataset, a widely-used collection of real-world text that served as a prominent language modeling benchmark until relatively recently.
>
> > the "general formula" is just a general formula, but there is no particular insight into how to choose the specific values
>
> The purpose of our paper is not to present a SOTA approach to learning from pairwise human preferences, but rather to provide a theoretical framework complementary to the RL framework to reason about that problem. The insight we provide on the general formula is that practitioners should pick f(x) and g(x) based on a) what they know about annotator behaviour (which is an empirical question) and b) how they would like the globally optimal model distribution to relate to the implicit preference distribution (for instance, be equal to it or be equal to a product of experts involving the preference distribution).
>
> Additionally, Theorem 2 provides a theoretical guarantee that, should the annotator behave according to a PBDE generative process for preferences (which encompasses the Luce choice rule, DPO, and a length-normalized Luce choice rule, among other processes), and should one be interested in modeling the annotator's implicit preference distribution directly, then using the same generative process in the model is the correct thing to do.
>
> > I don't understand the contribution [of section 4.1]
>
> The point of section 4.1 is that we can reinterpret "reward modeling" as modeling a probability distribution (the "implicit preference distribution"). The DPO paper does discuss the relationship between the Bradley-Terry model assumption and the logistic pairwise model, but always from the perspective of a "latent reward model". The DPO paper (and, to our best knowledge, the broader existing literature) does not consider the "latent reward" as a probability distribution to be modeled, but rather a quantity to be maximized under certain regularization constraints. In contrast, in section 4.1 we show that LHF can be formulated in purely probabilistic terms, without having to interpret the preference function as an RL reward function.
>
> > it is well-known that the KL is 0 if and only if the probability distributions are the same
>
> The fact that the KL is 0 if and only if the probability distributions are the same is not what Theorem 1 is about. There are two kinds of probability distributions to consider: the probability of the comparison outcome $p(y; x_A, x_B)$ and the probability $p^*(x)$ (implicit preference distribution) of or reward $r_\phi(x)$ associated with some observation $x$.
>
> The zero-KL argument pertains to $p(y; x_A, x_B)$ but says nothing about how $r_\phi(x)$ or $\pi_\theta(x)$ relate to $p^*(x)$ when the KL is zero for all possible $x_A$ and $x_B$ pairs. In fact, if our assumption on annotator behavior is wrong (e.g., if they behave according to a length-normalized Luce choice rule while we assume they behave according to the usual Luce choice rule) the KL-divergence being zero for all possible $x_A$ and $x_B$ pairs does _not_ result in the optimal learned reward or model being proportional to $p^*(x)$.

---

> > ### Author Response · Authors · 2024-01-10
> > **(continued)**
> >
> > > the authors could clarify what result they would expect in cases where there is no agreement between annotators
> >
> > If there is no agreement between annotators because they have different implicit preference distributions, we would expect the model to capture each annotator's preference distribution, as demonstrated in Appendix A.4 and Figure 11 in the Appendix. This requires to condition the model on a (discrete or continuous) latent variable as is done in Appendix A.4. Note that efficient and effective solutions to that remain an open problem; we do not claim to present such a solution outside the restricted setting of the univariate toy problem we investigate.
> >
> > > I also easily believe that in the case of "LLM alignment" where annotators are supposed to follow fairly strict guidelines such extreme disagreement where the middle ground is undesirable doesn't really happen.
> >
> > We are not aware of any existing work supporting the claim that extreme disagreements do not really happen. We also argue that human feedback is useful precisely in cases where defining strict and unambiguous guidelines is intractable.
> >
> > > this sentence seems inconsistent (high likelihood for sequences of low probability), maybe it's because the authors don't compute an actual likelihood (e.g., without taking into account the probability of "end of sequence" token)?
> >
> > We do take the probability of the EOS token into account; refer to the `TransformerLM` class in the supplementary material's notebook. Its call function sums over all log-likelihood vector elements associated with non-zero tokens (i.e., excludes the padding tokens), including the EOS token (which has value 2 in the code).
> >
> > We agree the statement seems contradictory because this is not an obvious phenomenon, but our claim is supported by Table 2. Specifically, we claim that the "long" model assigns higher likelihoods to individual short sequences relative to individual long sequences; we do not make any claims about log-likelihood magnitudes in an absolute sense. Note that because there are many more long sentences than short ones, it is possible to assign smaller likelihoods to each individual long sequence while still assigning a larger total probability mass to the set of long sequences.
> >
> > > the length bias is fairly obvious and the difference between sum log prob(token|history) vs mean log prob (token|history) is folklore
> >
> > We do not claim the length-normalized variant in Equation 26 as part of our contributions, and in fact we specifically cite Yuan et al. (2023)'s RRHF approach as an example of using length-normalization in practice. Our contribution is to provide an exposition to the problem through a clear and interpretable experiment.

---

### Review · Reviewer_mn8Y · 2024-01-08

**Summary Of Contributions:**

This paper investigates into the probabilistic perspective of reinforcement learning from human feedback. They showed that, assuming the generative process of human annotator and the model we trained share the same preference behavior distribution equation(PBDE) (or the so-called Luce choice rule as a special example), learning the global optimum of the loss function can be equivalent to learning the implicit preference distribution function (which is called by the authors a 'density estimation problem'). They also reviewed some alternative loss function for addressing the problem of learning from human preference. They also considered the problem of annotator misspecification and showed empirically that a mismatch between generative process and models can lead to a severely bad policy. They hence emphasize the importance of explicitly stating the assumptions of annotation generative process.

**Audience:**

Yes

**Broader Impact Concerns:**

/

**Claims And Evidence:**

Yes

**Requested Changes:**

See above.

**Strengths And Weaknesses:**

Strengths:

1. The authors formulate a new perspective of reinforcement learning from human feedback and show that under some assumptions, optimizing the loss function of RLHF is equivalent to learning the implicit preference distribution. From this perspective, it is possible to avoid fitting the reward function directly and learn from the preference pair directly.

2. They gave some simple examples in the annotation misspecification, which is easy to understand and emphasizes the importance of explicitly stating the assumptions on the annotation process. They also did sufficient experiments on learning the implicit preference distribution, and learning preference under the annotation misspecification.

3. The theory part seems solid and the proof is good o my knowledge.

Weaknesses:

1. The paper structure can be improved from my perspective. Here is my suggestions: you should give some motivations on why the Luce rule is unrealistic before section 4.3. The length-normalized variants (equation 26) will be a good example. In case that the context lengths vary in the dataset, there will be an implicit preference towards longer sentences. So a length-normalized variant of Luce rule is preferred and it belongs to the class of PBDE. Additionally, I am not sure what role the section 4.4 plays in your paper. Is it just a review for alternative loss functions in learning from human preference? If so and if this section does not connect very closely with your other parts, then probably this section can be moved to the appendix.

2. In section 5, you propose two generative models. You can successfully learn the 'density mixture' using a same model as the annotator, but if we want to learn the first type (the annotator mixture), how can we do it? In the appendix A.4, the authors proposed a way to deal with it, but to implement it, we need to know 'how many clusters are there in the annotation preference distribution.' In practice, how do we know it? More generally, you claimed that it is essential to explicitly state the assumption for the annotator behavior, but how can we know the annotator behavior if the annotators are some groups of people? (I know this question is kind of beyond the scope of this paper, but for the completeness of paper, I suggest at least the authors should discuss how people can get to know which assumption is true for the annotator behavior in existing literatures).

---

> ### Author Response · Authors · 2024-01-10
> **Response from the authors**
>
> Thank you for your constructive feedback!
>
> > you should give some motivations on why the Luce rule is unrealistic before section 4.3
>
> Following your suggestion, we moved the discussion earlier in the paper, right after Theorem 2 is introduced.
>
> > I am not sure what role the section 4.4 plays in your paper.
>
> You are correct that the section reviews alternative loss functions in learning from human preferences. We moved this section to the Appendix.
>
> > if we want to learn the first type (the annotator mixture), how can we do it? how can we know the annotator behavior if the annotators are some groups of people?
>
> These are excellent questions. We added a discussion in section 6 which we summarize here.
>
> Unsupervised discovery of annotator clusters is a difficult and open problem. As you rightfully point out, the naive approach we present in Appendix A.4 requires knowing the correct number of clusters in advance, which is a big drawback. The literature on probabilistic modeling suggests that we could perhaps replace the categorical random variable random for the cluster ID with a continuous random variable, but this has yet to be tried in practice.
>
> In some cases practitioners could conceivably use additional metadata to their advantage, for instance by assigning each annotator with a unique ID and solving the binary preference classification problem conditioned on the annotator ID. Alternatively, if practitioners know in advance that preferences are polarized and if the way each annotator leans can be predicted from certain covariates (age, location, etc.), those covariates could be used as additional context in the binary preference classification problem.
>
> As for how one can know whether they are making the correct assumption on annotator behavior, this is likely as difficult as determining whether the function family that a certain neural network architecture represents contains the true target function to approximate. Practitioners' decisions should be informed by results in disciplines like mathematical psychology or by their empirical knowledge. Additionally, should a model fail to learn from pairwise preferences, the possibility that a practitioner's behavioral assumptions are wrong should at least be considered.

---

### Author Response · Authors · 2024-01-10
**Manuscript update following the authors' response**

We would like to thank all reviewers again for their thoughtful and constructive feedback. We updated the manuscript taking this feedback into account and color-coded the changes we made in blue. We remain committed to addressing any additional feedback the reviewers may have.

---

### Decision · Action_Editor_1eap · 2024-02-20

**Recommendation:** Accept as is

**Comment:**

This paper presents a new perspective on RLHF. Specifically, it is shown that a generative process for preferences (the Luce choice rule) leads to a probabilistic interpretation of RLHF, namely as minimizing KL between the underlying human preferences and the distribution induced by the reward model.

Due to a split in the judgment of the reviewers, and owing to the uniqueness of the criteria for TMLR, I read through this paper somewhat carefully, and I am not sold on its value myself. Unlike other papers like DPO, the reformulation in this paper does not appear to buy us a whole lot, at least from the evidence presented here.

The paper emphasizes:
> your reward model secretly learns the annotator’s implicit preference distribution

I go back and forth on the value of this. It feels like a minor reformulation of the Bradley-Terry model, basically just making a generative assumption which doesn't dramatically change our interpretation of the existing procedure.

Moreover, as the paper itself notes,
> if one believes that the implicit preference distribution is the ideal generative model, then expressing the reward as the LLM’s log-likelihood is the correct thing to do

but this feels to me like somewhat circular reasoning: the Luce choice rule explains the algorithm already in use, and this paper doesn't really offer any techniques to evaluate whether or not the Luce choice rule is the right assumption to keep making.

The discussion of broader PBDEs and the reinterpretation of DPO is interesting. I'm again not quite sure how valuable this reformulation is. In particular, the theory in the DPO paper itself, showing the optimality of the DPO policy under the Bradley-Terry model, goes significantly beyond what is here in terms of its value.

The idea of dealing with annotator misspecification is interesting, but other work like rewarded soups ( https://arxiv.org/pdf/2306.04488.pdf ) and follow-ups push this further. The experiments presented here would unquestionably be valuable if they were the first forays in this area, but they are not. I am not sure about the experiments in section 5.2; while they deal with language modeling, the setting here is very toy.

My complaints here reflect the skepticisms of two reviewers who ended up being in favor of rejection. The objections were on the basis of technical novelty and the triviality of the theorems. I do feel that TMLR has some bar for this, represented by the "audience" bar. However, I am not confident to say that there are *no* individuals in the audience who would find this paper interesting. I think it is a thought-provoking piece of work that at least contributes to the conversation around these methods.  I also believe the edits help significantly in raising the technical sophistication of the discussion and in tying it to recently proposed methods.

In defense of the paper, I will say that it is technically correct (as far as I could tell) and quite well written. I found it enjoyable to read. I could see myself recommending it to students who want to gain some understanding of the formulations of these different methods and understand them from a new perspective. I also think this *could* be a platform for future work in this area; perhaps others in the audience besides myself and the critical reviewers will find significant value in what it presents!

**Audience:**

Yes, see below

**Claims And Evidence:**

Yes, see below